# NMR and SEM fractal dimensions explore shale pore structure taking the Upper Paleozoic shale in Ordos Basin as an example

**Keying Zhao**[ID]*, **Zhanghua Zhang**

Sichuan Water Conservancy Vocational College, Chengdu, China

* zhaokeyingdy@163.com

## Abstract

In this paper, the fractal dimension is calculated by extracting pore parameters from SEM images and NMR experimental data, the pore structure heterogeneity in plane and space is comprehensively discussed, and the relationship between the fractal dimension and shale composition and physical parameters is discussed, providing new ideas for the study of shale reservoirs heterogeneity. Fractal dimension analysis of SEM images reveals that the shale pores of the Shanxi Formation can be divided into organic pores, inter-granular pores and micro-fractures. The average diameter of nano-scale pores is 17.13 nm to 67.65 nm, the surface porosity is 5.75% to 9.37%, and the proportion of micro-fractures is 0.36% to 0.72%, with an average value of 0.53%. The ImageJ Weka Segmentation module in ImageJ software intelligently optimizes the degree of pore recognition in SEM images to ensure accurate extraction and characterization of pore structure features. The fractal dimension of the SEM image was calculated using the Dathe formula for the identified pores: Fractal dimension of bound fluid pore (0.4922～0.9396) and fractal dimension of movable fluid (2.9727～2.989), quartz content has a negative correlation with the fractal dimension of bound fluid pores, clay mineral content has a positive correlation with the fractal dimension of bound fluid pores, NMR fractal dimension has no obvious correlation with organic matter content and maturity, and NMR fractal dimension has a negative correlation with porosity, but has no obvious correlation with permeability: indicating that NMR fractal dimension is mainly affected by the composition of shale minerals; The Shanxi Formation shale has a high degree of evolution but the organic matter pores are not developed. The reservoirs space is mainly provided by inter-granular pores and micro-fractures; the inter-granular pores and micro-fractures have high heterogeneity and poor connectivity leads to low permeability.This paper attempts to use the ImageJ Weka Segmentation module to intelligently optimize the identification of pores, which improves the efficiency and accuracy of pore identification. At the same time, it combines the fractal dimension of SEM images and the fractal

**Data availability statement:** All relevant data are within the paper and its Supporting Information files.

**Funding:** The presented work was funded by Zhao Keying's scientific research start-up fund, which is provided by Sichuan Water Conservancy Vocational College, and the fund number is 2024030401. The funders had no role in study design, data collection and analysis, decision to publish, or preparation of the manuscript.

**Competing interests:** The authors have declared that no competing interests exist.

dimension of NMR images to characterize reservoir characteristics, which provides a basis for quantitatively describing the irregularity of shale pore morphology.

## 1 Introduction

Shale reservoirs have significant microscopic heterogeneity, which has a great impact on the storage, enrichment and migration of shale gas [1]. Previous people used rock sample observation, high-pressure mercury injection experiments and other methods to analyze the heterogeneity characteristics of pores, fractures, and mineral components of shale reservoirs [1–3]. A characterization and evaluation model for shale pore heterogeneity was established based on evaluation parameters such as pore size ratio and coefficient of variation [4–8]. Since the porous media spaces and pore interfaces of shale pores both have fractal structures, fractal geometry is an effective tool to study the heterogeneity of pore structure in shale reservoirs [9–11]. Fractal dimensions can be used to quantitatively characterize the heterogeneity of reservoir pore structure. and complexity, establishing a bridge connecting the reservoir microscopic pore structure (pore size distribution and connectivity) and the macro reservoir physical properties (porosity, permeability and adsorption) [12–13]. Therefore, studying reservoir pore structure and fractal characteristics is of great significance for evaluating shale reservoirs.

Currently, the main methods to obtain the fractal dimension of shale pores include: SEM image analysis [14–15], nitrogen adsorption model [16], mercury porosimetry [17], thin slices [18], FIB-SEM [19] and multifractal theory based on 3D X-ray micro-computed tomography images [20], etc.. Tang et al. used a nitrogen adsorption model combined with field emission scanning electron microscopy (FE-SEM) to study the fractal dimension characteristics of shale pores with different evolution degrees [21]. SEM image analysis is a means to directly observe pore morphology and structure, and NMR experiments are non-destructive, fast and accurate. Therefore, this paper combines these two methods to study the fractal and pore structure characteristics of shale reservoirs in the Shanxi Formation of the Upper Paleozoic in the Ordos Basin. The fractal dimension is calculated by extracting pore parameters from SEM images and NMR experimental data. The heterogeneity of pore structure in plane and space is comprehensively studied, and the relationship between fractal dimension and shale components and physical properties parameters is discussed, thus providing new ideas for studying the heterogeneity of shale reservoirs.

In recent years, drilling results in coal-series formations of the Carboniferous and Permian of the Upper Paleozoic in the Ordos Basin have shown that, except for a small amount of low-yield shale gas flow obtained after fracturing and transformation of some exploration wells, most exploration wells are limited to "good gas logging shows","high gas volume or gas content","flammable ignition or successful ignition", etc., and no substantial breakthroughs in shale gas development have been achieved, among which reservoir conditions are an important constraint. In order to further clarify the pore characteristics and controlling factors of the Upper Paleozoic shale in the Ordos Basin, this paper takes samples from drilling and field sections in the northeastern Ordos Basin as examples, combines SEM images and NMR to characterize the reservoir

characteristics, and quantitatively describes the irregular degree of shale pore morphology from the perspective of fractal dimension, providing theoretical support for predicting favorable areas of organic-rich shale gas reservoirs.

## 2 General situation of regional geology and experimental samples

Marine-continental transitional shale is an important field of oil and gas exploration in China. Carboniferous-Permian marine-continental transitional shale is one of the three mechanism-rich shales in China. The drilling and well test results around sea-land transitional shale gas reveal that it has good gas-bearing properties and development prospects [22]. The Upper Paleozoic in Ordos Basin is a marine-continental transitional facies, where develop several sets of shale rich with organic matter, and the shale of Shanxi formation is the most developed. Except for the northern Yimeng uplift, the thickness of shale is more than 30 meters. The thickness of the shale to the south of the Wushen Banner-Shenmu-Fugu line in the Yishan slope belt is more than 40 meters.Seven vertical wells in Daning-Jixian area have been tested by fracture fracturing of shale in Shanxi formation. Five wells have obtained industrial gas flow, and the highest test production is more than $1 \times 10^4 m^3/d$ [22], indicating that the Upper Paleozoic shale has great development potential. In this experiment, the shale samples of the Upper Paleozoic Shanxi formation in the Ordos Basin are analyzed and tested, and the study area is located in the location shown in the blue box in Fig 1.

## 3 Analysis and test methods

### 3.1 Organic geochemical analysis and petrophysical parameters

The experimental instrument of total organic carbon is LECO CS-200 C030 analyzer, the detection method is the determination method of total organic carbon in sedimentary rock (GB/T19145-2003), the pyrolysis experimental instrument

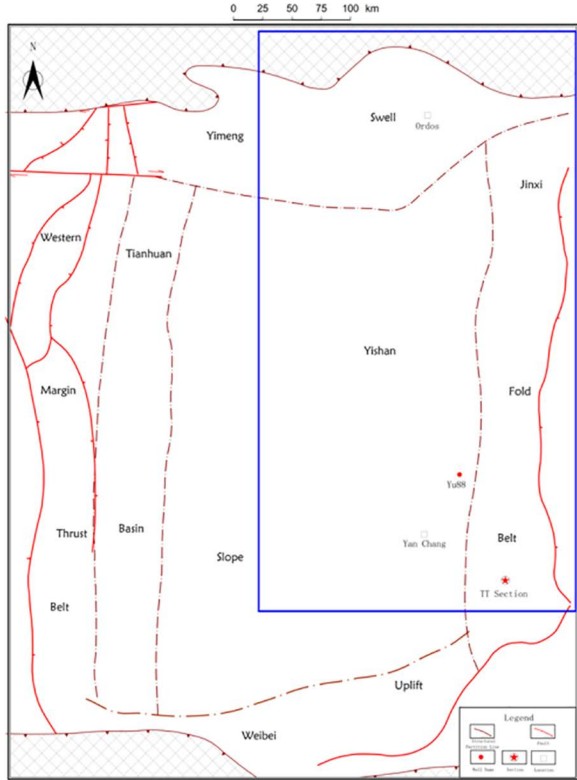

**Fig 1. Location map of the study area.**

is MPV- III analyzer, the detection method is vitrinite reflectance measurement method in sedimentary rock (SY/T5124-2012), the X-ray diffraction experimental instrument is D8 DISCOVER X-ray diffractometer, the detection is based on SY/T 5163–2010.

### 3.2 NMR experiment

Nuclear magnetic resonance experimental measurement of rock samples is the use of experimental instruments to measure the nuclear magnetic relaxation time of rock samples. According to the measured values of 1 (longitudinal relaxation time) and 2 (transverse relaxation time), the parameters such as porosity, permeability, type of producing fluid, free fluid index and irreducible water saturation which are less affected by lithology are calculated. The NMR experiments in this study were conducted using a RecCore-04 low-field NMR rock sample analyzer. The parameter measurement standards followed were SY/T 6490–2000, and the sample centrifugation pressure was 200 psi.

### 3.3 SEM images

After the sample was polished by LJB- 1A ion thinning instrument, the surface of the prepared rock sample was ground by argon ion for 2 hours on the Hitachi IM4000 equipment, and the uneven parts and surface attachments were removed. The SEM images obtained can directly provide information such as pore morphology and pore structure.

## 4 Fractal dimension of SEM images

### 4.1 Pore classification

The results of SEM analysis show that the shale pores of the Upper Paleozoic in the study area can be divided into organic matter pores [Fig 2a, 2c], inter-granular pores [Fig 2e, 2g] and micro-fractures (including marginal shrinkage fractures of organic matter) [Fig 2l, 2k, 2m, 2o]. According to the size, it can be divided into nano-scaled pores (pore diameter < 1 μ m), micron pores (1mm ~ 1 μ m) and micro-cracks, the average pore size of nano-scaled pores is 17.13nm ~ 67.65nm, the average surface porosity is 5.75% ~ 9.37%, the average pore size of micron pores is 3.17μm ~ 5.61μm, the average surface porosity is 10.2% ~ 19.32%, and the percentage of micro-cracks is 0.36% ~ 0.72%, with an average of 0.53%.

### 4.2 Optimization of pore extraction from SEM images

The extraction and optimization of image pores is completed by ImageJ Weka Segmentation software. By classifying and identifying a small number of substrates and pores on the image, the software can separate the substrates and pores of the entire image by learning the given substrate and pore data. The more data you give, the more accurate the software will recognize it. The process is as follows.

The image of SEM is processed by Image J software, and the pores and fractures in shale can be extracted [Fig 2b, 2d, 2f, 2h, 2j, 2l, 2n, 2p]. When extracting pores, Image J software needs to set a grayscale threshold for the image. The adjustment of the grayscale value of the image is affected by the operator's experience and image resolution, and there is often confusion between pores and matrix: the position that is not a pore is identified as a pore [the position indicated by the red circle in Fig 3b, 3c, 3d]. The location of the original pore is identified as the matrix [the position indicated by the green circle in Fig 3b], and the area of the pore is identified too large [the position indicated by the blue circle in Fig 3e and 3h], resulting in insufficient accuracy of pore calculation. In this paper, intelligent optimization tool is introduced to optimize the identification of pores and matrix in a picture: (1) import an image into the ImageJ software to determine the scale of the picture; (2) copy a picture (which is the same size as the original image) and cut out the scale mark below the picture; (3) Open the ImageJ Weka Segmentation intelligent optimization tool and import the pictures into the tool; (4) apply the recognition tool to mark the pores and matrix on the picture, and then export. (5) when the distribution of pores

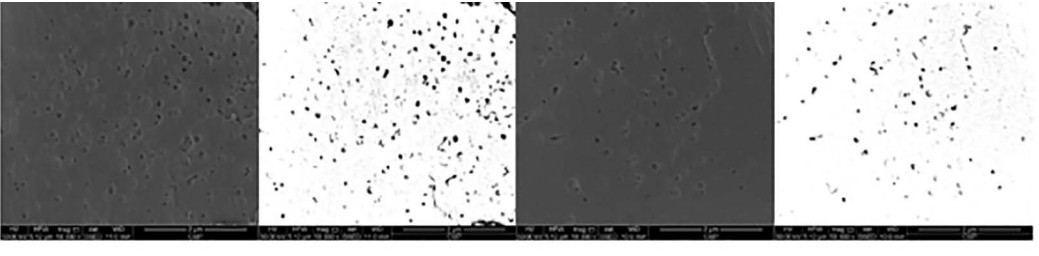

(a) TT-1 Organic matter porosity  (b) TT-1 Organic matter pore extraction (c) TT-2Organic matter porosity (d) TT-2 Organic matter pore extraction

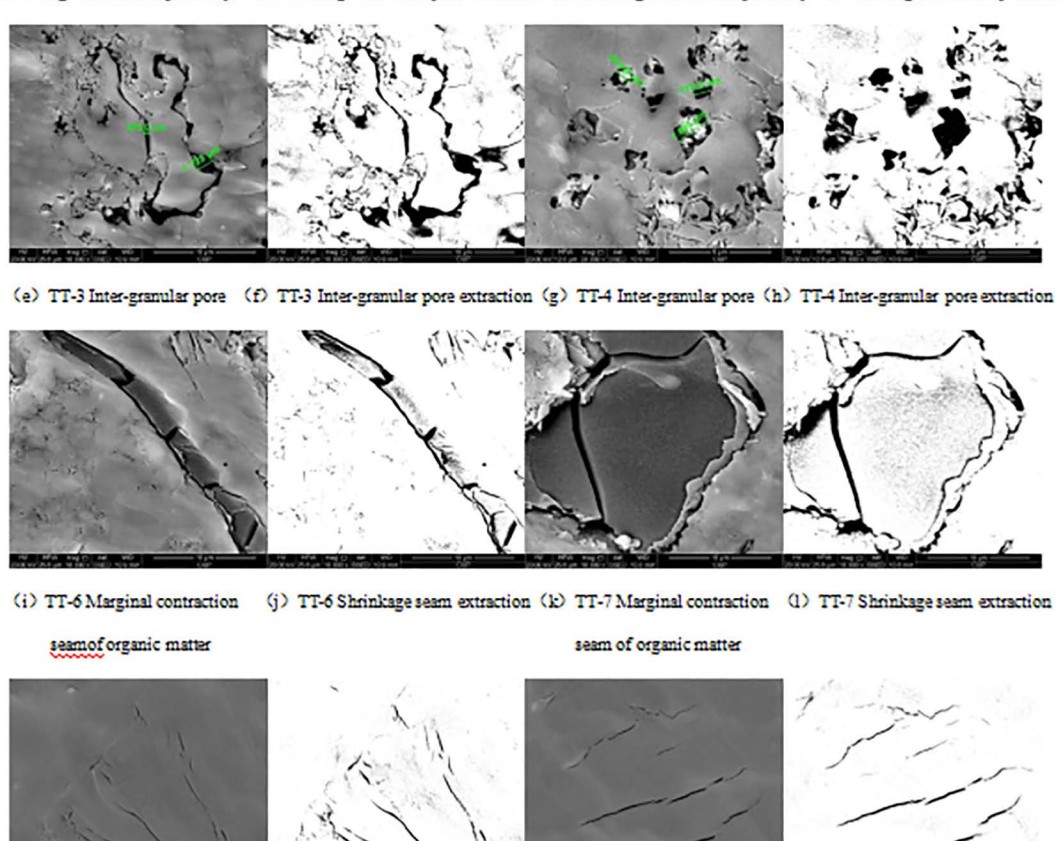

(e) TT-3 Inter-granular pore    (f) TT-3 Inter-granular pore extraction  (g) TT-4 Inter-granular pore  (h) TT-4 Inter-granular pore extraction

(i) TT-6 Marginal contraction    (j) TT-6 Shrinkage seam extraction  (k) TT-7 Marginal contraction  (l) TT-7 Shrinkage seam extraction

seam of organic matter                                                                      seam of organic matter

(m) Y88-1 Micro crack      (n) Y88-1 Micro-crack extraction      (o) Y88-2 Micro crack      (p) Y88-2Micro-crack extraction

**Fig 2. Pore types in SEM images.**

and matrix in the first derived picture is quite different from that of the original image, repeat step (4) until the distribution map of pores or fractures consistent with the original image is obtained [Fig 3c, 3f, 3l].

The image optimized and recognized by the ImageJ Weka Segmentation intelligent recognition module is then subjected to pore structure analysis: set the image to 8-bit and perform binarization processing. The binarization processing uses ImageJ's default iteration method. The image is segmented into two parts by assuming a threshold T. The average values M1 and M2 are calculated for each part. Assuming that T '=(M1+M2) is not equal to T, then T=T' is made and then iterative continues until the two are equal. Then, the line edges in the image are closed, the pores are closed, and the

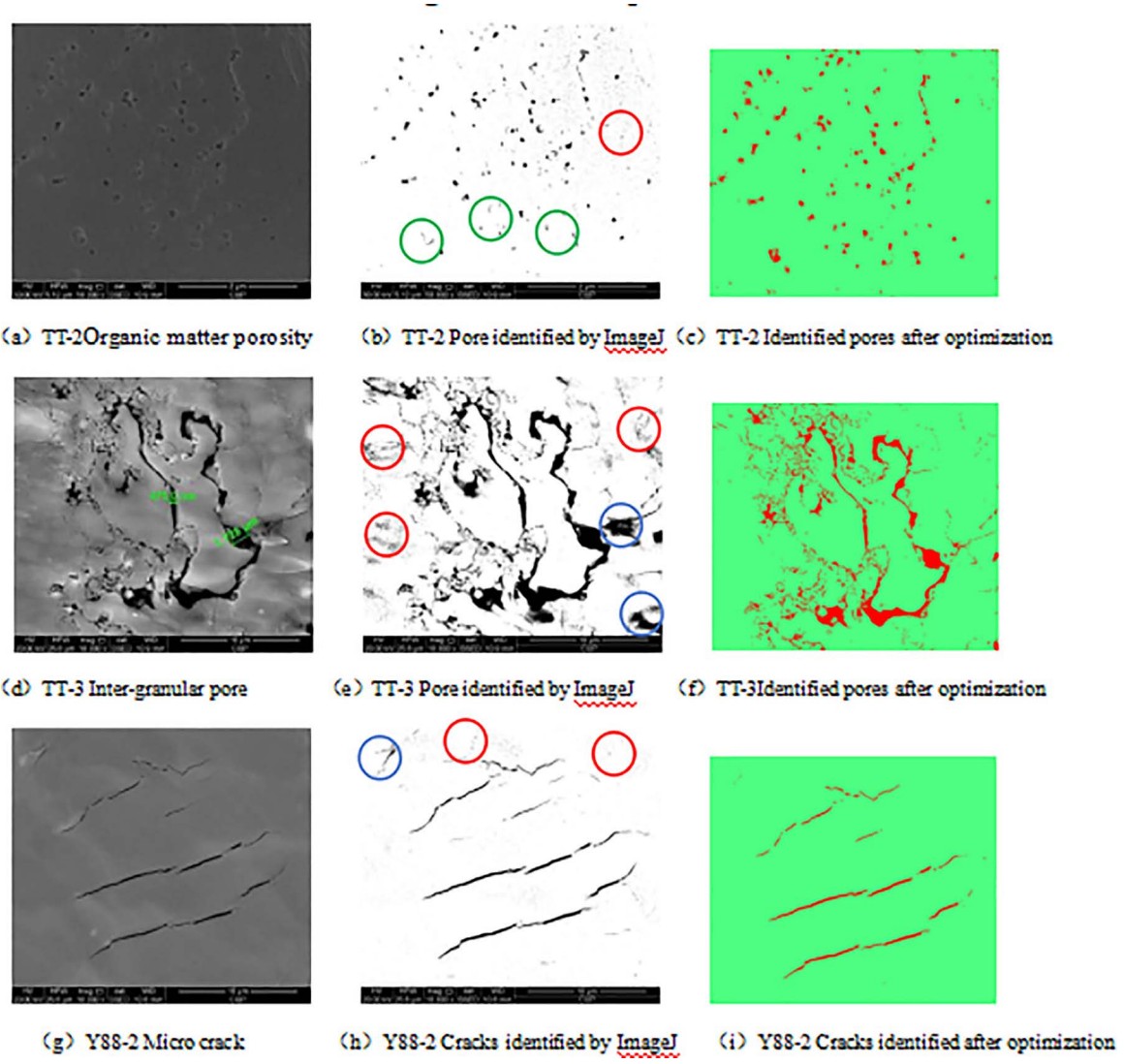

(a) TT-2 Organic matter porosity (b) TT-2 Pore identified by ImageJ (c) TT-2 Identified pores after optimization

(d) TT-3 Inter-granular pore (e) TT-3 Pore identified by ImageJ (f) TT-3 Identified pores after optimization

(g) Y88-2 Micro crack (h) Y88-2 Cracks identified by ImageJ (i) Y88-2 Cracks identified after optimization

**Fig 3. Comparison of extracted pores in SEM images and optimized pores.**

noise is removed. Then, software analysis functions are applied to obtain parameters such as pore area, circumference, and face ratio [23]. The optimized image using Image J Weka Segmentation intelligent identification has very high consistency with the pores in the original image, and the calculated pores can better reflect the true state of shale pores.

### 4.3 Calculation of fractal dimension of SEM images

Although SEM images can visually reflect the shape and size of pores, there is no legal quantitative representation of the irregular degree of pore morphology. In this paper, fractal dimension is used to analyze the irregular degree of shale pores.

Refer to Dathe's [14] formula for calculating fractal dimension of two-dimensional plane image:

$$\lg C = D_i/2 \cdot \lg S + C_t \tag{1}$$

In the formula: $C$ is the pore perimeter; $S$ is the pore area; $D_i$ is the fractal dimension of a certain pore; $C_t$ is the coefficient.

It can be seen from the formula of Dathe et al. that if there is an obvious linear relationship between $\lg C$ and $\lg S$, the fractal dimension of the SEM image is twice the slope, and the fractal dimension of the SEM image can be obtained.

### 4.4 Image result analysis

In the optimized picture, the area and perimeter of pores and cracks are calculated by Image J, and the fractal dimensions of different pores and micro-cracks are obtained by logarithmic fitting of area and perimeter (Fig 4, Table 1). Graphically, there is an obvious linear relationship between the logarithm of area and perimeter, and the fitting rate is 98%. The fractal dimension of Upper Paleozoic shale is between 1.16 and 1.5, and the fractal dimensions of different pore types (organic pores, inter-granular pores and micro-fractures) are different (Fig 4). The statistical results show that the fractal dimension of organic matter pores are the smallest (1.16), the fractal dimension of inter-granular pores are the second (1.28-1.36), and that of fractures are the highest (1.34-1.51). The main reason for this phenomenon is that the pores of organic matter are undeveloped, round, and the roundness is relatively high, the circumference is smooth and the fractal dimension is lower; the complexity of inter-granular pores and fracture edge lines leads to higher fractal dimension.

## 5 NMR fractal dimension

### 5.1 Determine the threshold value($T_{2c}$)

The $T_2$ spectra of the same sample were measured before and after centrifugation. Before centrifugation, that is, when the rock is saturated with water, a curve varying with relaxation time can be obtained. By adding the relaxation amplitudes at different time points after normalization, a cumulative curve varying with the relaxation time can be obtained. With the same treatment of $T_2$ spectrum after centrifugation, the same $T_2$ spectrum curve and cumulative $T_2$ spectrum curve varying with relaxation time can be obtained. The difference of the maximum amplitude of $T_2$ spectrum before and after centrifugation is the percentage of effective porosity to the total porosity. The relaxation time corresponding to the point where the horizontal extension of $T_2$ spectrum after centrifugation intersects with that of $T_2$ spectrum before centrifugation is $T_{2c}$ (Fig 5). When the relaxation time is less than the relaxation time threshold value, it is the bound fluid in the residual pores, and when the relaxation time is greater than the relaxation time threshold value, it is the movable fluid in the effective pores. According to the calculation, the relaxation time threshold value of Shanxi formation shale of Upper Paleozoic in Ordos Basin is 1.28~2.8ms.

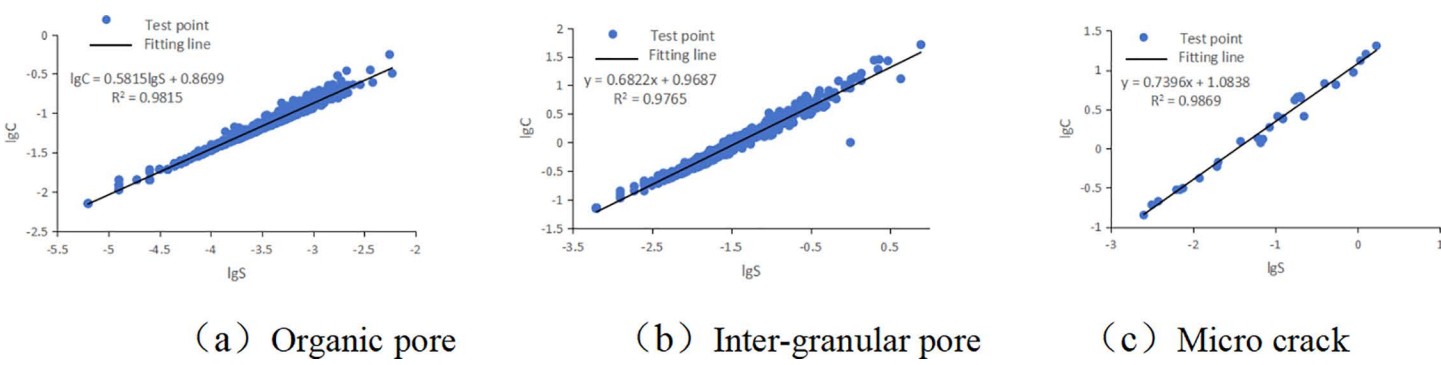

Fig 4. Fractal dimension of SEM image.

**Table 1. Tables of different pore parameters based on SEM images.**

| Sample number | Pore type | Average area/nm² | Average perimeter/nm | Fitting formula | Shape factor | Decision coefficient | Fractal dimension |
|---|---|---|---|---|---|---|---|
| TT37–18 | Organic pore | 2989.25 | 203.84 | lg C = 0.5830lg S + 0.8475 | 0.583 | 0.9789 | 1.166 |
| TT08–22 | Organic pore | 417.18 | 72.01 | lg C = 0.5815lg S + 0.8699 | 0.5815 | 0.9815 | 1.163 |
| TT38–12 | Inter-granular pore | 131776.01 | 1798.92 | lg C = 0.6651lg S + 0.9280 | 0.6651 | 0.9858 | 1.3302 |
| TT08–09 | Inter-granular pore | 94520.94 | 1348.30 | lg C = 0.6822lg S + 0.9687 | 0.6822 | 0.9765 | 1.3644 |
| Y88-05 | Inter-granular pore | 14671.62 | 410.39 | lg C = 0.6443lg S + 0.9756 | 0.6443 | 0.9847 | 1.2886 |
| Y88-11–09 | Inter-granular pore | 213508.16 | 2031.59 | lg C = 0.6634lg S + 0.9171 | 0.6634 | 0.9871 | 1.3268 |
| Y88-21 | Inter-granular pore | 107329.78 | 1164.05 | lg C = 0.6559lg S + 0.9108 | 0.6559 | 0.9833 | 1.3118 |
| Y88-24 | Inter-granular pore | 188180.21 | 1435.73 | lg C = 0.6460lg S + 0.8831 | 0.646 | 0.9815 | 1.292 |
| Y88-36 | Inter-granular pore | 57704.47 | 1125.39 | lg C = 0.6726lg S + 1.0201 | 0.6726 | 0.9826 | 1.3452 |
| Y88-38 | Inter-granular pore | 415682.59 | 2652.00 | lg C = 0.6621lg S + 0.8958 | 0.6621 | 0.9845 | 1.3242 |
| TT08-04-1 | Micro crack | 1015375.0 | 4744.75 | lg C = 0.6714lg S + 0.8386 | 0.6714 | 0.9269 | 1.3428 |
| TT08–17 | Micro crack | 91881.54 | 1833.93 | lg C = 0.7313lg S + 1.1188 | 0.6799 | 0.9823 | 1.3598 |
| Y88-09–16 | Micro crack | 259576.72 | 3792.44 | lg C = 0.7396lg S + 1.0838 | 0.7396 | 0.9869 | 1.4792 |
| Y88-11–11 | Micro crack | 80993.60 | 1770.81 | lg C = 0.7546lg S + 1.1435 | 0.7546 | 0.9866 | 1.5092 |
| TT08–17 | Micro crack | 271569.67 | 2710.35 | lg C = 0.6799lg S + 0.9516 | 0.7313 | 0.9811 | 1.4626 |

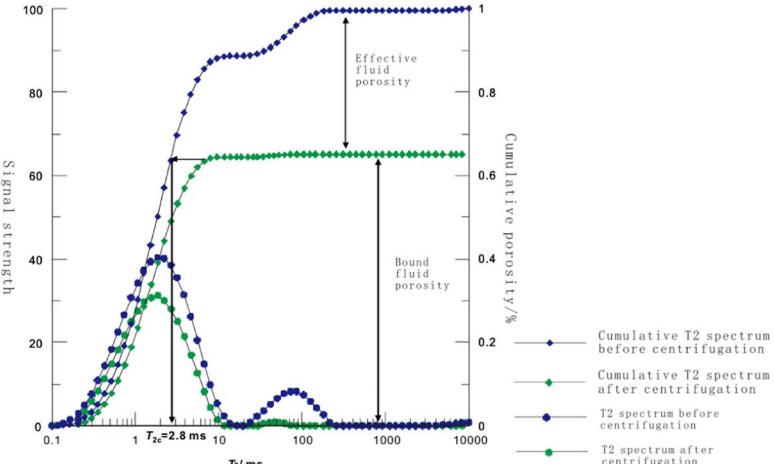

**Fig 5. Distribution of residual, effective porosity and threshold value.**

## 5.2 Calculation method of NMR Fractal Dimension

The experimental sample of NMR is cylindrical with diameter 38.1mm and length 25mm~50mm. The present research shows that the more complex the pore structure, the stronger the heterogeneity and the higher the fractal dimension [16、21、24、25]. The fractal dimension of NMR has been widely used in recent years. At present, the formula proposed by ZHOU [26–27] is mainly used to calculate it. According to the fractal theory of geometry, the fractal geometric approximation equation corresponding to NMR $T_2$ spectrum is [28–29].

$$S_v = (T_{2\,max}/T_2)^{D-3}$$

(2)

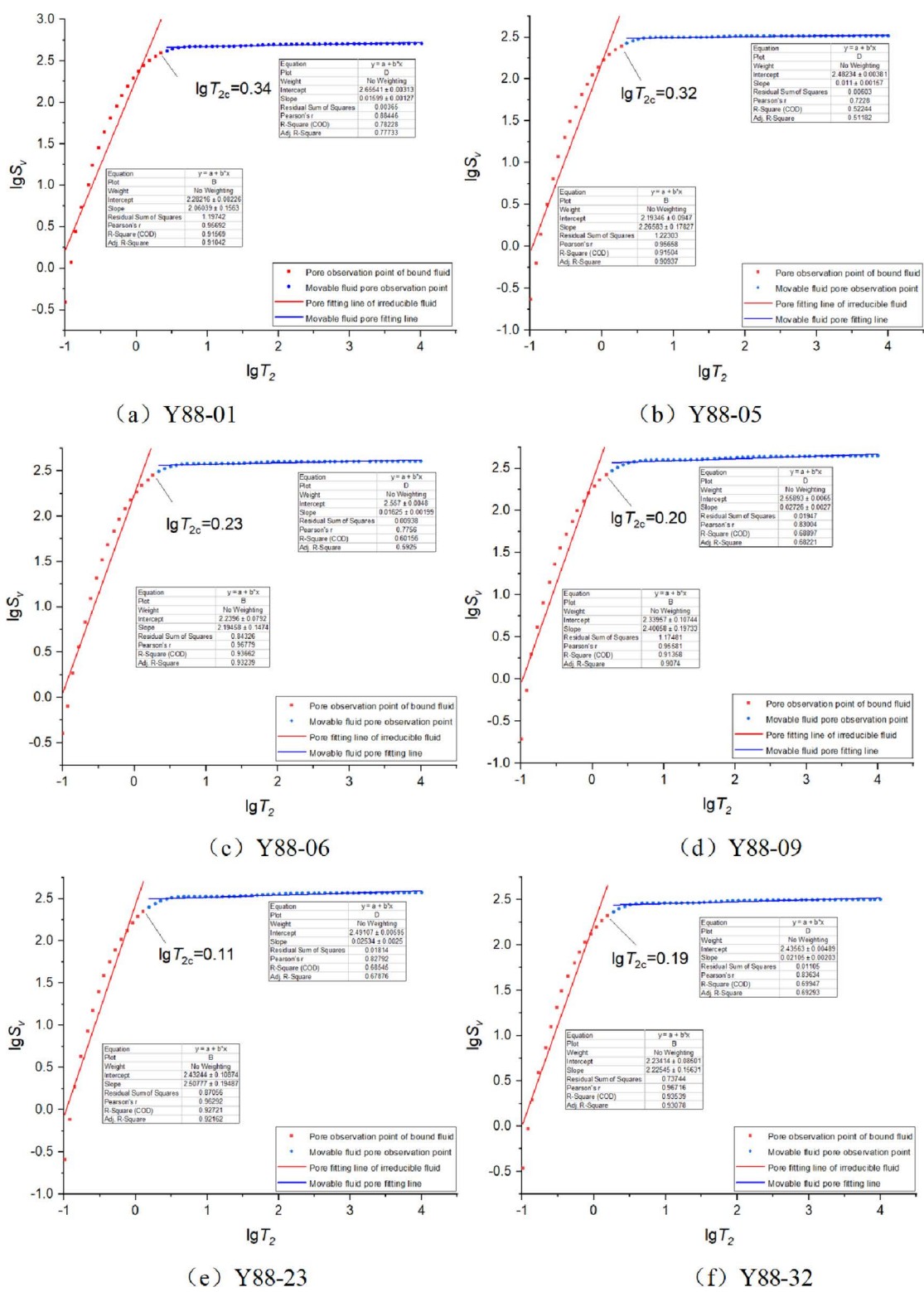

**Fig 6. NMR fractal dimension curve.**

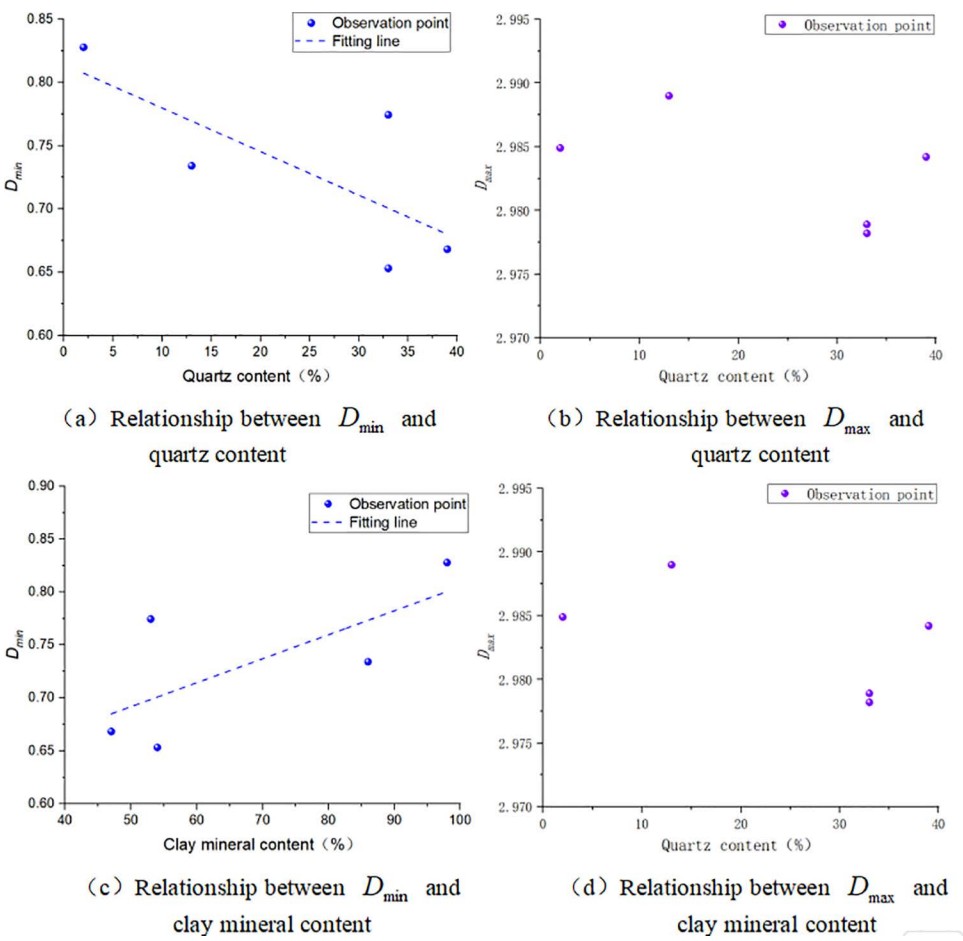

（a）Relationship between $D_{\min}$ and quartz content

（b）Relationship between $D_{\max}$ and quartz content

（c）Relationship between $D_{\min}$ and clay mineral content

（d）Relationship between $D_{\max}$ and clay mineral content

**Fig 7. Relationship between NMR fractal dimension and mineral content of quartz and clay.**

The logarithm on both sides can be deduced as

$$\lg S_v = (3 - D)\lg T_2 + (D - 3)\lg T_{2\,\max} \tag{3}$$

In the formula: $S_v$ is the percentage of cumulative pore volume to the total pore volume when the relaxation time is less than $T_2$; D is the fractal dimension; $T_{2\,\max}$ is the maximum relaxation time.

If the shale pores have self-similar pore structure and fractal characteristics, then formula (3) shows a linear relationship.

## 5.3　Obtaining fractal dimensions in sections

The fractal dimension often has obvious segmented characteristics, and most researchers fit the parting curve by segments [9、30]. The relaxation time threshold value is an effective dividing line for identifying movable fluid pores and irreducible fluid pores. In this paper, the relaxation time valve point logarithm is used as the demarcation point, and the $\lg S_v$ and $\lg T_2$ scatter point diagrams are divided into two different stages. The points larger than the relaxation time valve point logarithm and less than the relaxation time valve point logarithm are linearly fitted. According to the fitting curve, the fractal dimension of NMR is obtained, $D_{\min}$ is the pore fractal dimension of bound fluid, and $D_{\max}$ is the pore fractal dimension of

**Table 2. Statistical results of NMR fractal dimension.**

| Sample Number | $T_2 < T_{2c}$ | | | | $T_2 > T_{2c}$ | | | |
|---|---|---|---|---|---|---|---|---|
| | NMR regression equation | $D_{min}$ | Slope | Decision coefficient | NMR regression equation | $D_{max}$ | Slope | Decision coefficient |
| Y88-01 | $\lg S_v = 2.0604 \lg T_2 + 2.2822$ | 0.9396 | 2.0604 | 0.9157 | $\lg S_v = 0.0160 \lg T_2 + 2.6554$ | 2.984 | 0.016 | 0.7823 |
| Y88-06 | $\lg S_v = 2.1946 \lg T_2 + 2.2396$ | 0.8054 | 2.1946 | 0.9367 | $\lg S_v = 0.0163 \lg T_2 + 2.557$ | 2.9837 | 0.0163 | 0.6216 |
| Y88-23 | $\lg S_v = 2.5078 \lg T_2 + 2.4324$ | 0.4922 | 2.5078 | 0.9272 | $\lg S_v = 0.0253 \lg T_2 + 2.4911$ | 2.9747 | 0.0253 | 0.6855 |
| Y88-32 | $\lg S_v = 2.2255 \lg T_2 + 2.2341$ | 0.7745 | 2.2255 | 0.9354 | $\lg S_v = 0.0211 \lg T_2 + 2.4356$ | 2.9789 | 0.0211 | 0.6995 |
| Y88-35 | $\lg S_v = 2.3468 \lg T_2 + 2.3173$ | 0.6532 | 2.3468 | 0.9136 | $\lg S_v = 0.0218 \lg T_2 + 2.4823$ | 2.9782 | 0.0218 | 0.6880 |
| Y88-37 | $\lg S_v = 2.3317 \lg T_2 + 2.3334$ | 0.6683 | 2.3317 | 0.9489 | $\lg S_v = 0.0158 \lg T_2 + 2.4383$ | 2.9842 | 0.0158 | 0.5772 |
| Y88-05 | $\lg S_v = 2.2658 \lg T_2 + 2.1935$ | 0.7342 | 2.2658 | 0.9150 | $\lg S_v = 0.0110 \lg T_2 + 2.4823$ | 2.989 | 0.0110 | 0.5224 |
| Y88-09 | $\lg S_v = 2.4006 \lg T_2 + 2.3396$ | 0.5994 | 2.4006 | 0.9136 | $\lg S_v = 0.0273 \lg T_2 + 2.5589$ | 2.9727 | 0.0273 | 0.6890 |
| Y88-38 | $\lg S_v = 2.1721 \lg T_2 + 2.2335$ | 0.8279 | 2.1721 | 0.9246 | $\lg S_v = 0.0151 \lg T_2 + 2.5367$ | 2.9849 | 0.0151 | 0.6560 |

movable fluid [31]. The slope and fractal dimension can be calculated by linear fitting. It can be seen from Table 2 that the fractal dimension of bound fluid is 0.4922~0.9396, with an average of 0.7216, and the pore fractal dimension of movable fluid is 2.9727~2.989, with an average of 2.9811. NMR fractal dimension curve (Fig 6) shows that the linear fitting correlation coefficient of most samples is more than 0.6 (Table 2), indicating that the pore distribution is self-similar and can be characterized by fractal theory.

## 6 Discussion

### 6.1 Relationship between NMR fractal dimension and shale composition

The relationship between quartz content and NMR fractal dimension shows that there is a negative correlation between pore fractal dimension of bound fluid and quartz content [Fig 7a], but there is no obvious correlation between pore fractal dimension of movable fluid and quartz content [Fig 7b]. The reasons may be that quartz pores are underdeveloped, contributing minimally to the reservoir space. If quartz micropores contain fluid, which makes the rough surface homogeneous, the fractal dimension may be reduced [32]. The surface of quartz is relatively smooth, and the increase of quartz content weakens the degree of pore heterogeneity and irregularity, thus reducing the pore fractal dimension. The results are different from the experimental results of Zhou et al on the shale of NiuTiTang formation in northern Guizhou [26], The content of quartz in marine shale is positively correlated with total organic carbon, and its sedimentary environment is conducive to the enrichment and preservation of organic matter and the development of authigenic quartz, so there is a positive correlation between them. In addition, due to the lack of quartz-related pores, the specific surface area and pore volume provided by quartz are limited, and the fractal dimension is positively correlated with pore specific surface area and pore volume [32]. Therefore, the pore fractal dimension of bound fluid is negatively correlated with the content of quartz.

JI et al show that the clay mineral content is closely related to the specific surface area, but not to the pore volume [32]. This is mainly due to the layered structure of clay minerals, which is stacked between layers, thus increasing the specific surface area, but for the pore volume of the cake is smaller [25,32]. The relationship between clay mineral content and NMR fractal dimension of Upper Paleozoic marine shale in Ordos basin shows that there is a positive correlation between pore fractal dimension of bound fluid and clay mineral content [Fig 7c], but there is no obvious correlation between pore fractal dimension of movable fluid and clay mineral content [Fig 7d]. Sun and other studies show that different clay mineral content has a great influence on the specific surface area [33]. The adsorption capacity of clay minerals from large to small is montmorillonite, illite/smectite interlaying, kaolinite, chlorite and Illite [34]. Montmorillonite has not been detected in

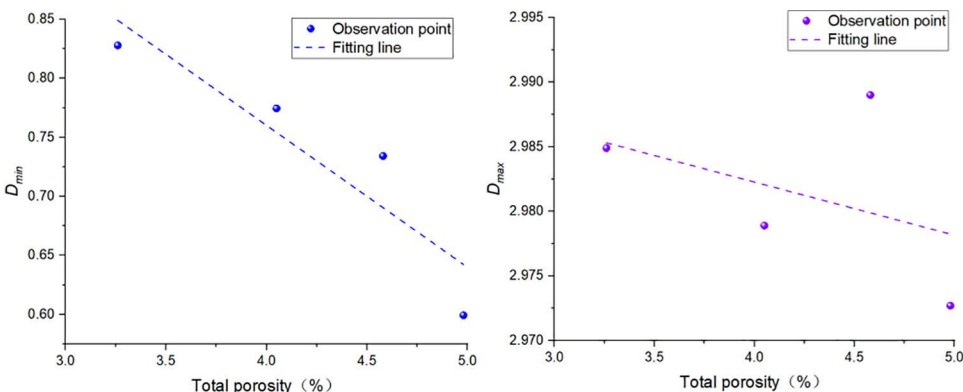

**Fig 8. Relationship between NMR fractal dimension and total porosity.**

the upper Paleozoic shale in the study area, the proportion of illite/smectite interlaying and kaolinite is between 58% and 99%, and the content of chlorite and Illite is less, which leads to the roughness of mineral surface and strong adsorption capacity and the pore fractal dimension of bound fluid increases.

The correlation between total organic carbon content, Ro and NMR fractal dimension of Upper Paleozoic shale in Ordos Basin is not obvious. Organic matter pores are rare in shale, only 2 samples in argon ion-polished samples show a small number of organic matter pores, which are not interconnected. This indicates that the contribution of organic matter pores to movable fluid pores and bound fluid pores is minimal, so the organic carbon content and NMR fractal dimension is not closely related. Zhou et al [35] show that maturity has a positive correlation with pore fractal dimension of irreducible fluid, but no obvious correlation with pore fractal dimension of movable fluid, and the correlation coefficient between total organic carbon and pore fractal dimension of bound fluid is higher than that of maturity. This shows that compared with maturity, total organic carbon has a great influence on the pore fractal dimension of movable fluid, and with the increase of total organic carbon, the pore volume and specific surface area increase. The difference between the experimental results of this study and Zhou may be due to the fact that the two parameters of maturity and fractal dimension are affected by many factors: on the one hand, the maturity of Upper Paleozoic shale is too high, and the Ro is more than 3.0%, which has reached the over-mature stage. Although there are many organic pores in hydrocarbon generation of organic matter, there are differences in hydrocarbon generation rate and conversion rate among different kerogen types, and the effect of maturity is not obvious. On the other hand, although maturity can promote the shrinkage of organic matter and produce inter-granular pores between organic matter and brittle minerals, due to the large buried depth, inter-granular pores are often closed by mechanical compaction and cementation of brittle particles [36], resulting in no close relationship between maturity and NMR fractal dimension.

## 6.2 Relationship between NMR Fractal Dimension and Shale physical Properties

There is a negative correlation between bound fluid pores, movable fluid pores and total porosity (Fig 8), and there is no obvious correlation between bound fluid pores, movable fluid pores and permeability. The smaller the fractal dimension in the sample is, the weaker the pore heterogeneity is, the smaller the specific surface area of the pore is and the larger the average pore diameter is, which can provide more pore space [32]. The development of larger pores will reduce the complexity of specific surface area and pore distribution, resulting in a decrease in the fractal dimension of movable fluid pores. In addition, the development of smaller pores in shale can also increase porosity to a certain extent, resulting in a negative correlation between porosity and pore fractal dimension of irreducible fluid. Permeability is usually affected by comprehensive factors such as pore roar, porosity, roundness and pore connectivity [37]. The relationship between NMR

fractal dimension and permeability is not obvious, which is mainly due to the poor permeability of the upper Paleozoic shale pores in the Ordos basin, and the proportion of connected roar volume in the pores is only 0.85%-0.92%, so that the movable fluid occupies less pores and the pore permeability is lower, only in 0.00023md~0.00054md.

## 7 Conclusion

(1) The pore heterogeneity of Upper Paleozoic shale in Ordos basin is strong, and the pore size varies greatly. The fractal dimension of SEM image is between 1.16 and 1.51. The fractal dimensions of different types of pores are obviously different, the organic matter is the lowest, the inter-granular pore is the second, and the fracture is the highest.

(2) The NMR fractal dimension has two stages, namely the pore fractal dimension of bound fluid ($D_{min}$) and the fractal dimension of movable fluid ($D_{max}$). The NMR fractal dimension has a linear correlation with the main mineral composition of shale, but no obvious correlation with organic carbon content and Ro, indicating that mineral composition plays an important role in the development of pores. The fractal dimensions of bound fluid pores and movable fluid pores are negatively correlated with total porosity and have no obvious correlation with permeability, indicating that the heterogeneity of pores is strong and the connectivity of pores is poor.

(3) The fractal dimension of SEM image can be used to reflect the diversity of pore morphology and the development of different pore types in shale, and the fractal dimension of NMR can be used to evaluate the quality of shale reservoirs. The application of fractal dimension can effectively analyze the pore structure and morphology of shale and provide a new idea for the evaluation of shale reservoirs quality and the study of heterogeneity.

## Supporting information

**S1 File. Supporting information.**
(RAR)

## Acknowledgments

Authors want to thank to professor Guo Shaobin (School of Energy Resources, China University of Geosciences (Beijing), Beijing, China) for the analytical data.

## Author contributions

**Data curation:** Zhanghua Zhang.

**Methodology:** keying zhao.

**Software:** keying zhao.

**Writing – original draft:** keying zhao.

**Writing – review & editing:** keying zhao.

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
