## [Decision Letter · Decision Letter 0]

6 Nov 2024

PONE-D-24-42271Discussion on pore characteristics of shale based on NMR and SEM fractal dimension

-- taking the Upper Paleozoic shale in Ordos Basin as an examplePLOS ONE

Dear Dr. zhao,

Thank you for submitting your manuscript to PLOS ONE. After careful consideration, we feel that it has merit but does not fully meet PLOS ONE’s publication criteria as it currently stands. Therefore, we invite you to submit a revised version of the manuscript that addresses the points raised during the review process.

We look forward to receiving your revised manuscript.

Kind regards,

Fateh Bouchaala

Academic Editor

PLOS ONE

Journal Requirements:

2. We note that this submission includes NMR spectroscopy data. We would recommend that you include the following information in your methods section or as Supporting Information files:

1) The make/source of the NMR instrument used in your study, as well as the magnetic field strength. For each individual experiment, please also list: the nucleus being measured; the sample concentration; the solvent in which the sample is dissolved and if solvent signal suppression was used; the reference standard and the temperature.

2) A list of the chemical shifts for all compounds characterised by NMR spectroscopy, specifying, where relevant: the chemical shift (δ), the multiplicity and the coupling constants (in Hz), for the appropriate nuclei used for assignment.

3)The full integrated NMR spectrum, clearly labelled with the compound name and chemical structure.

We also strongly encourage authors to provide primary NMR data files, in particular for new compounds which have not been characterised in the existing literature. Authors should provide the acquisition data, FID files and processing parameters for each experiment, clearly labelled with the compound name and identifier, as well as a structure file for each provided dataset. See our list of recommended repositories here: https://journals.plos.org/plosone/s/recommended-repositories

4. Thank you for stating the following financial disclosure: The presented work was funded by Zhao Keying's scientific research start-up fund, which is provided by Sichuan Water Conservancy Vocational College, and the fund number is 2024030401. 

5. We note that your Data Availability Statement is currently as follows: All relevant data are within the manuscript and its Supporting Information files.

Additional Editor Comments:

The authors should highlight the main contribution of this work to the topic of fractals or/and to the investigation of the Upper Paleozoic Shanxi Formation of the Ordos Basin. Furthermore, introduction should be re-written and English editing is needed.

Best regards

Reviewers' comments:

Reviewer's Responses to Questions

**Comments to the Author**

1. Is the manuscript technically sound, and do the data support the conclusions?

Reviewer #1: Yes

Reviewer #2: Yes

2. Has the statistical analysis been performed appropriately and rigorously? 

Reviewer #1: Yes

Reviewer #2: No

3. Have the authors made all data underlying the findings in their manuscript fully available?

Reviewer #1: Yes

Reviewer #2: Yes

4. Is the manuscript presented in an intelligible fashion and written in standard English?

Reviewer #1: Yes

Reviewer #2: No

5. Review Comments to the Author

Reviewer #1: In this paper, authors proposed to examine the pore structure and fractal characteristics of shale in the Upper Paleozoic Shanxi Formation of the Ordos Basin. The study analyses the relationship between the NMR fractal dimension and factors such as mineral composition, geochemical parameters, and physical properties, utilizing SEM imaging and artificial intelligence techniques.

The writing quality of the paper is of a moderate level, so I recommend authors to improve the general quality of the writing based on the comments hereafter especially for the introduction which is quite weak.

Technically, results in this study are interesting but I do not see novelty in the paper, and I urge authors to emphasize the novelty in the approach as I could not see it clearly. Moreover, I have several questions and suggestions to improve the paper content and quality as at this stage. They can be found in the attached file.

Reviewer #2: The manuscript does not fully adhere to the required format. For example, it uses "0. Introduction," "1. ~," and "2. ~" instead of following standard headings. Also, the line spacing on pages 12 and 15 is inconsistent, likely due to the use of MathType for in-line equations. Consider replacing these with Microsoft Word’s built-in equation editor (“Insert > Equation > Insert New Equation”). Additionally, a PDF format of the manuscript should be submitted to ensure the format remains stable during submission.

In Chapter 1, it would be beneficial to include a well location map, regional map, or geological map to illustrate the geological setting as Figure 1. I also recommend adding photos of the experimental equipment as Figures 2a, 2b, 2c, and so forth.

In Chapter 2.2, the first sentence repeats “Nuclear magnetic resonance.” Please revise to avoid redundancy.

In Figure 2(c) and (d), the blue circle you mention is missing. Please correct the figure or the associated text.

In Equation 1, you introduce , but do not define the role of . Please clarify this in the text.

Figure 4 contains four plots with arrows and descriptions that overlap with the plots. To improve readability, place the legend separately on either the right or left side of the figure.

In general, many sentences in the manuscript are overly long and complex, likely due to an attempt to condense information. For example, on the bottom of page 18: “Organic matter pores are rare in shale; only 2 samples in argon ion-polished samples have organic matter pores and a small number of pores are developed. Organic matter pores account for a small proportion of organic matter and usually cannot be connected. Organic matter pores contribute minimally to movable fluid pores and irreducible fluid pores, so the relationship between organic carbon content and NMR fractal dimension is weak.” Please revise to keep sentences shorter and clearer throughout the manuscript.

Finally, in Figure 7b, you note a negative correlation between movable fluid pores and total porosity. However, the y-axis range (2.970–2.995) is inconsistent with the previous range (2.978–2.990). Please present Figure 7b with a y-axis range of 2.978–2.990, along with the correlation coefficient value.

6. PLOS authors have the option to publish the peer review history of their article (what does this mean? ). If published, this will include your full peer review and any attached files.

**Do you want your identity to be public for this peer review?** For information about this choice, including consent withdrawal, please see our Privacy Policy .

Reviewer #1: **Yes: ** Mohamed Soufiane Jouini

Reviewer #2: No

---

## [Author Response · Author response to Decision Letter 1]

15 Dec 2024

Dear Editor:

These are the revisions made to the manuscript based on the review comments.

For Comment 1: The format of the article has been revised.

For Comment 2 (1): The model of the nuclear magnetic resonance (NMR) spectrometer and the magnetic field strength have been added. All samples were tested using the same instrument and under the same pressure conditions. Since the samples are shale, there is no information on cell nuclei; sample concentration; solvent used for dissolving the samples and whether solvent signal suppression was applied; reference standards and temperature, etc. (2) (3) The NMR testing of shale focuses on measuring the water and gas saturation after centrifugation, analyzing the development degree of shale pores through the difference in saturation before and after centrifugation, without detecting compound information. I will submit the NMR test report to the database. For Comment 3: The research content of this paper originates from my doctoral dissertation, so no permits were required. For Comment 4: The funders had no role in study design, data collection and analysis, decision to publish, or preparation of the manuscript. The research start-up funding provided economic support for me to conduct this part of the research at this institution. For Comment 5: Additional data has been supplemented and uploaded to PACE. The images were generated by software, which the PACE system does not accept.

Response to Reviewer 1's Comments:

Added maps showing the location of the study area, well positions, and profiles.

1.The advantage of FIB-SEM experiments lies in their capability for microfabrication of materials. SEM experiments are primarily used for observing surface morphology and compositional analysis of materials but do not possess microfabrication capabilities. This study mainly focuses on observing the pore structure characteristics of rock samples, thus SEM experimental analysis was adopted.

2.Provided a brief introduction to the method of smart pore recognition using ImageJ Weka Segmentation and described its advantages.

3.Revised the sentences that needed modification as suggested by the reviewer. We are very grateful for the valuable comments provided by the reviewer.

4.Made comprehensive revisions to the abstract of the paper.

Response to Reviewer 2's Comments:

1.Revised the format of the paper according to the publication standards.

2.Added regional geological maps.

3.Modified the content of the paper, removing redundant parts. Text corresponding to Figures 3(c) and (d) has been revised. Described the function of Formula 1. Revised Figure 5. Unified the scale of the y-axis for Figures 7 and 8. We are very grateful for the valuable comments provided by the reviewer.

---

## [Decision Letter · Decision Letter 1]

29 Dec 2024

PONE-D-24-42271R1NMR and SEM fractal dimensions explore shale pore structure

—— taking the Upper Paleozoic shale in Ordos Basin as an examplePLOS ONE

Dear Dr. zhao,

Thank you for submitting your manuscript to PLOS ONE. After careful consideration, we feel that it has merit but does not fully meet PLOS ONE’s publication criteria as it currently stands. Therefore, we invite you to submit a revised version of the manuscript that addresses the points raised during the review process.

We look forward to receiving your revised manuscript.

Kind regards,

Fateh Bouchaala

Academic Editor

PLOS ONE

Additional Editor Comments:

Comments and questions from one of the reviewers were not addressed. Please address them to improve the manuscript and make it publishable.

Reviewers' comments:

Reviewer's Responses to Questions

**Comments to the Author**

1. If the authors have adequately addressed your comments raised in a previous round of review and you feel that this manuscript is now acceptable for publication, you may indicate that here to bypass the “Comments to the Author” section, enter your conflict of interest statement in the “Confidential to Editor” section, and submit your "Accept" recommendation.

Reviewer #1: (No Response)

Reviewer #2: All comments have been addressed

2. Is the manuscript technically sound, and do the data support the conclusions?

Reviewer #1: Partly

Reviewer #2: Yes

3. Has the statistical analysis been performed appropriately and rigorously? 

Reviewer #1: Yes

Reviewer #2: Yes

4. Have the authors made all data underlying the findings in their manuscript fully available?

Reviewer #1: No

Reviewer #2: Yes

5. Is the manuscript presented in an intelligible fashion and written in standard English?

Reviewer #1: No

Reviewer #2: Yes

6. Review Comments to the Author

Reviewer #1: In this paper, authors proposed to examine the pore structure and fractal characteristics of shale in the Upper Paleozoic Shanxi Formation of the Ordos Basin. The study analyses the relationship between the NMR fractal dimension and factors such as mineral composition, geochemical parameters, and physical properties, utilizing SEM imaging and artificial intelligence techniques.

Several of my comments stated in my first revision were ignored and left without response. Authors should either implement suggested improvements or reply why they refuse to implement them. They cannot ignore reviewers comments it is not acceptable. Also, the writing quality of the paper still need significant improvement.

Introduction:

The introduction is too short and does not highlight enough the work done in fractals to characterize rock properties. It need a significant re-writing.

• Authors did not highlighted the novelty in the approach as I requested in my first revision.

• 1 introduction -> Introduction

• shale reservoir -> Shale reservoir

• “At present, the fractal dimension is mainly obtained by SEM image analysis”

This statement is not true and should be corrected indeed several research studies used FIB-SEM, 3D X-Ray Micro-Computed. Please see the list of papers that you should refer to in your paper.

- Sulieman H, Jouini MS, Alsuwaidi M, Al-Shalabi EW, Al Jallad OA (2024) Multiscale investigation of pore structure heterogeneity in carbonate rocks using digital imaging and SCAL measurements: A case study from Upper Jurassic limestones, Abu Dhabi, UAE. PLoS ONE 19(2): e0295192. https://doi.org/10.1371/journal.pone.0295192

- Jouini, M.S., Alabere, A.O., Alsuwaidi, M. et al. Experimental and digital investigations of heterogeneity in lower cretaceous carbonate reservoir using fractal and multifractal concepts, (2023). Scientific Reports 13, 20306 https://doi.org/10.1038/s41598-023-47681-w

• Section 4.2: “The picture optimized by ImageJ Weka Segmentation artificial intelligence is then analyzed by Image J software: set the picture to 8-bit, carry on binary processing, close the edge of the line in the picture, close the pore, fill the hole, and apply the software analysis function to get the parameters such as pore area, perimeter, area porosity and so on[20]” Authors stated that they open the Weka Segmentation artificial intelligence tool and use it to improve the image segmentation results. However they did not specify which AI method was used what is its sensitivity advantages and limitations. Authors need to state this as it is a crucial step of the method proposed.

• Section 4.3 Traditionally, the usual formula used for Dathe's formula is based on box-counting technique to compute the fractal dimension Df:

N(ϵ) is the number of boxes of side length ϵ that are needed to cover the structure and ϵ represents the box size.

The equation provided in (1) is not common so please provide references at least to know who proposed it and in which context it was used.

• Several authors tried to investigate possible relationship between experimental fractal dimension (here NMR) and image fractal dimension (here SEM). Since you have all the results could you provide any comments regarding this?

Reviewer #2: The advantage of FIB-SEM experiments lies in their capability for microfabrication of materials.

SEM experiments are primarily used for observing surface morphology and compositional analysis

of materials but do not possess microfabrication capabilities. This study mainly focuses on observing

the pore structure characteristics of rock samples, thus SEM experimental analysis was adopted.

Provided a brief introduction to the method of smart pore recognition using ImageJ Weka

Segmentation and described its advantages.

Revised the sentences that needed modification as suggested by the reviewer. We are very grateful

for the valuable comments provided by the reviewer.

Made comprehensive revisions to the abstract of the paper.

7. PLOS authors have the option to publish the peer review history of their article (what does this mean? ). If published, this will include your full peer review and any attached files.

**Do you want your identity to be public for this peer review?** For information about this choice, including consent withdrawal, please see our Privacy Policy .

Reviewer #1: **Yes: ** Mohamed Soufiane Jouini

Reviewer #2: No

---

## [Author Response · Author response to Decision Letter 2]

23 Feb 2025

To Reviewers

1. The introduction of the paper was revised.

2. The abstract of the paper is revised and the innovations of this paper are highlighted

3. Reviewer's "1 introduction" and "shale reservoirs""At present, the fractal dimension is mainly obtained by SEM image analysis" have been revised.

4.Image J Weka Segmentation is an intelligent identification module in ImageJ software. It was originally used for cell identification in medicine, but used in this paper to identify the pore structure of shale.

5.The fractal dimension of SEM images adopts the perimeter-area method, which is used in "Fractal Method for Estimating Porosity","Calculation of Fractal Dimension of Soil","Fractal Dimension Measured from Area-Perimeter Relationship and Material Toughness" and other articles. Fractal dimension method is a method used to describe and analyze complex shapes and structures. Since this method can study the fractal dimension of soil, urban boundary dimension and other aspects, it can of course also be used to study the characteristics of shale pores.Dathe's The Surface Fractal Dimension of Soil-pore Interface as Measured by Image Analysis. The pore area and perimeter in the SEM image in this paper can be calculated by ImageJ software, so this method is used to calculate the fractal dimension of the pore.Therefore, this paper quotes the perimeter-area method used in Dathe's paper to calculate the pore fractal dimension in SEM images.

---

## [Editor Report · Decision Letter 2]

5 Mar 2025

PONE-D-24-42271R2NMR and SEM fractal dimensions explore shale pore structure—— taking the Upper Paleozoic shale in Ordos Basin as an examplePLOS ONE

Dear Dr. Zhao,

Thank you for submitting your manuscript to PLOS ONE. After careful consideration, we feel that it has merit but does not fully meet PLOS ONE’s publication criteria as it currently stands. Therefore, we invite you to submit a revised version of the manuscript that addresses the points raised during the review process.

We look forward to receiving your revised manuscript.

Kind regards,

Fateh Bouchaala

Academic Editor

PLOS ONE

Additional Editor Comments:

Dear authors,

Many relevant comments from one of the reviewers were not addressed. I invite the authors to fully address these comments before submitting the next revision.

Regards

---

## [Author Response · Author response to Decision Letter 3]

7 Mar 2025

The article has been revised twice, and the following is a reply to the content of each revision.

The first time was as follows:

Response to Dear Editor's Comments

These are the revisions made to the manuscript based on the review comments.

For Comment 1: The format of the article has been revised.

For Comment 2 (1): The model of the nuclear magnetic resonance (NMR) spectrometer and the magnetic field strength have been added. All samples were tested using the same instrument and under the same pressure conditions. Since the samples are shale, there is no information on cell nuclei; sample concentration; solvent used for dissolving the samples and whether solvent signal suppression was applied; reference standards and temperature, etc. (2) (3) The NMR testing of shale focuses on measuring the water and gas saturation after centrifugation, analyzing the development degree of shale pores through the difference in saturation before and after centrifugation, without detecting compound information. I will submit the NMR test report to the database. For Comment 3: The research content of this paper originates from my doctoral dissertation, so no permits were required. For Comment 4: The funders had no role in study design, data collection and analysis, decision to publish, or preparation of the manuscript. The research start-up funding provided economic support for me to conduct this part of the research at this institution. For Comment 5: Additional data has been supplemented and uploaded to PACE. The images were generated by software, which the PACE system does not accept.

Response to Reviewer 1's Comments

1.Added maps showing the location of the study area, well positions, and profiles.

2.The advantage of FIB-SEM experiments lies in their capability for microfabrication of materials. SEM experiments are primarily used for observing surface morphology and compositional analysis of materials but do not possess microfabrication capabilities. This study mainly focuses on observing the pore structure characteristics of rock samples, thus SEM experimental analysis was adopted.

3.Provided a brief introduction to the method of smart pore recognition using ImageJ Weka Segmentation and described its advantages.

4.Revised the sentences that needed modification as suggested by the reviewer.

5.Made comprehensive revisions to the abstract of the paper.

We are very grateful for the valuable comments provided by the reviewer.

Response to Reviewer 2's Comments:

1.Revised the format of the paper according to the publication standards.

2.Added regional geological maps.

3.Modified the content of the paper, removing redundant parts. Text corresponding to Figures 3(c) and (d) has been revised. Described the function of Formula 1. Revised Figure 5. Unified the scale of the y-axis for Figures 7 and 8.

We are very grateful for the valuable comments provided by the reviewer.

The second time is as follows

To Reviewers

1. The introduction of the paper was revised.

2. The abstract of the paper is revised and the innovations of this paper are highlighted

3. Reviewer's "1 introduction" and "shale reservoirs""At present, the fractal dimension is mainly obtained by SEM image analysis" have been revised.

4.Image J Weka Segmentation is an intelligent identification module in ImageJ software. It was originally used for cell identification in medicine, but used in this paper to identify the pore structure of shale.

5.The fractal dimension of SEM images adopts the perimeter-area method, which is used in "Fractal Method for Estimating Porosity","Calculation of Fractal Dimension of Soil","Fractal Dimension Measured from Area-Perimeter Relationship and Material Toughness" and other articles. Fractal dimension method is a method used to describe and analyze complex shapes and structures. Since this method can study the fractal dimension of soil, urban boundary dimension and other aspects, it can of course also be used to study the characteristics of shale pores.Dathe's The Surface Fractal Dimension of Soil-pore Interface as Measured by Image Analysis. The pore area and perimeter in the SEM image in this paper can be calculated by ImageJ software, so this method is used to calculate the fractal dimension of the pore.Therefore, this paper quotes the perimeter-area method used in Dathe's paper to calculate the pore fractal dimension in SEM images.

---

## [Decision Letter · Decision Letter 3]

4 Apr 2025

PONE-D-24-42271R3NMR and SEM fractal dimensions explore shale pore structure—— taking the Upper Paleozoic shale in Ordos Basin as an examplePLOS ONE

Dear Dr. zhao,

Thank you for submitting your manuscript to PLOS ONE. After careful consideration, we feel that it has merit but does not fully meet PLOS ONE’s publication criteria as it currently stands. Therefore, we invite you to submit a revised version of the manuscript that addresses the points raised during the review process.

We look forward to receiving your revised manuscript.

Kind regards,

Fateh Bouchaala

Academic Editor

PLOS ONE

Journal Requirements:

Additional Editor Comments:

I urge the authors to carefully review the reviewers' comments, which have not been addressed since first revised version. The reviewers' comments are reasonable and can be adequately addressed.

Reviewers' comments:

Reviewer's Responses to Questions

**Comments to the Author**

1. If the authors have adequately addressed your comments raised in a previous round of review and you feel that this manuscript is now acceptable for publication, you may indicate that here to bypass the “Comments to the Author” section, enter your conflict of interest statement in the “Confidential to Editor” section, and submit your "Accept" recommendation.

Reviewer #1: (No Response)

2. Is the manuscript technically sound, and do the data support the conclusions?

Reviewer #1: Yes

3. Has the statistical analysis been performed appropriately and rigorously? 

Reviewer #1: Yes

4. Have the authors made all data underlying the findings in their manuscript fully available?

Reviewer #1: Yes

5. Is the manuscript presented in an intelligible fashion and written in standard English?

Reviewer #1: Yes

6. Review Comments to the Author

Reviewer #1: Introduction:

- Authors stated: “Currently, there are three main methods to obtain the fractal

dimension of shale pores: SEM image analysis [14-15], nitrogen adsorption model [16] and mercury porosimetry [17].”

This statement is not true and should be corrected indeed several research studies used digital images such as thin section, FIB-SEM, 3D X-Ray Micro-Computed in addition to other experimental laboratory measurements such as NMR.

Please see the list of papers that you should refer to in your paper.

- D. Abdassah, P. Permadi, R. Sumantri. Saturation exponents derived from fractal modeling of thin sections Presented at the SPE Asia Pacific Oil and Gas Conference Held in Adelaide, Australia (1996)

- Z. Zhang, A. Weller Fractal dimension of pore-space geometry of an Eocene sandstone formation Geophysics, 79 (6) (2014), pp. 377-387

- - Sulieman H, Jouini MS, Alsuwaidi M, Al-Shalabi EW, Al Jallad OA (2024) Multiscale investigation of pore structure heterogeneity in carbonate rocks using digital imaging and SCAL measurements: A case study from Upper Jurassic limestones, Abu Dhabi, UAE. PLoS ONE 19(2): e0295192. https://doi.org/10.1371/journal.pone.0295192

- Jouini, M.S., Alabere, A.O., Alsuwaidi, M. et al. Experimental and digital investigations of heterogeneity in lower cretaceous carbonate reservoir using fractal and multifractal concepts, (2023). Scientific Reports 13, 20306 https://doi.org/10.1038/s41598-023-47681-w

Section 4.1:

Authors stated the following:” The images optimized and identified by the ImageJ Weka Segmentation intelligent recognition module are then analyzed for pore structure: set the picture to 8-bit, carry on binary processing”.

I understand that authors used software to process the segmentation and geometric characterization of pores. However, using the software should be detailed in terms of the methodology used. In my previous revision I requested to authors to explain which segmentation method is sued to “carry on binary processing”. The authors did not address this request.

7. PLOS authors have the option to publish the peer review history of their article (what does this mean? ). If published, this will include your full peer review and any attached files.

**Do you want your identity to be public for this peer review?** For information about this choice, including consent withdrawal, please see our Privacy Policy .

Reviewer #1: **Yes: ** Mohamed Soufiane Jouini

---

## [Author Response · Author response to Decision Letter 4]

10 Apr 2025

Dear reviewer

Thank you for your constructive comments on my manuscript.We agree with the suggestions and will incorporate the recommended changes into the manuscript.

1.Based on Reviewer #1, after reading the references given by Reviewer, the statement "Authors stated: " Currently, there are three main methods to obtain the fractional "is revised to “Currently, the main methods to obtain the fractal dimension of shale pores include: SEM image analysis [14-15], nitrogen adsorption model [16], mercury porosimetry [17], thin slices [18], FIB-SEM[19] and multifractal theory based on 3D X-ray micro-computed tomography images [20], etc..” (see revised pages 2). References have been revised simultaneously(see revised pages 17).

2.The binary processing method is explained, and the default iteration method of Image J is adopted: The image is segmented into two parts by assuming a threshold T. The average values M1 and M2 are calculated for each part. Assuming that T '=(M1+M2) is not equal to T, then T=T' is made and then iterative continues until the two are equal(see revised pages 6-7).

We sincerely appreciate the time and effort invested by the reviewers in evaluating our manuscript. We look forward to any additional feedback or suggestions.

---

## [Editor Report · Decision Letter 4]

17 Apr 2025

NMR and SEM fractal dimensions explore shale pore structure

—— taking the Upper Paleozoic shale in Ordos Basin as an example

PONE-D-24-42271R4

Dear Dr. Keying,

We’re pleased to inform you that your manuscript has been judged scientifically suitable for publication and will be formally accepted for publication once it meets all outstanding technical requirements.

Kind regards,

Fateh Bouchaala

Academic Editor

PLOS ONE

Additional Editor Comments (optional):

The authors improved the manuscript after multiple revisions, by addressing authors suggestions.
---

## [Editor Report · Acceptance letter]

PONE-D-24-42271R4

PLOS ONE

Dear Dr. zhao,

I'm pleased to inform you that your manuscript has been deemed suitable for publication in PLOS ONE. Congratulations! Your manuscript is now being handed over to our production team.

Kind regards,

on behalf of

Dr. Fateh Bouchaala

Academic Editor

PLOS ONE